# Evaluation of Wear on Primary Tooth Enamel and Fracture Resistance of Esthetic Pediatric Crowns Manufactured from Different Materials

**DOI:** 10.3390/medicina60101678

**Published:** 2024-10-13

**Authors:** Nagehan Aktaş, Merve Bankoğlu Güngör

**Affiliations:** 1Department of Pediatric Dentistry, Faculty of Dentistry, Gazi University, 06490 Ankara, Türkiye; 2Department of Prosthetic Dentistry, Faculty of Dentistry, Gazi University, 06490 Ankara, Türkiye; mervegungor@gazi.edu.tr

**Keywords:** 3D-printing, CAD-CAM, fracture strength, primary tooth, tooth wears

## Abstract

*Background and Objectives:* Advances in dental materials and CAD-CAM technology have expanded crown options in primary teeth due to their improved appearance and mechanical properties. Thus, this study aimed to assess the enamel wear and fracture resistance of prefabricated, milled, and 3D-printed esthetic pediatric crowns. *Materials and Methods:* The study involved 60 extracted maxillary second primary molars and 60 3D-printed resin dies, divided into six groups based on different crown materials (n = 10): prefabricated zirconia, prefabricated composite, milled composite, milled resin matrix ceramic, milled PEEK, and 3D-printed resin. Prefabricated crowns were selected after the preparation of the typodont mandibular second primary molar tooth, while milled and 3D-printed crowns were custom produced. The specimens underwent mechanical loading of 50 N at 1.6 Hz for 250,000 cycles with simultaneous thermal cycling. The 3D and 2D wear amounts were evaluated by scanning the specimens before and after aging. Then, the fracture resistance and failure types of the restorations were recorded. *Results:* The results showed that the milled PEEK group had superior fracture resistance compared to the other groups, while prefabricated zirconia crown group had the lowest value. Milled resin matrix ceramic crown group displayed the lowest 3D wear volume, while 3D-printed crown group showed the highest 2D wear. *Conclusions:* The restorative material type did not have a significant effect on the wear of primary tooth enamel. The fracture resistance of the tested materials differed according to the material type. Although the milled PEEK group showed the highest fracture resistance, all tested materials can withstand chewing forces in children.

## 1. Introduction

The restoration of primary teeth is fundamental in pediatric dentistry for maintaining space, ensuring proper chewing, and guiding the eruption of permanent teeth. Failure to restore damaged or decayed primary teeth can result in early tooth loss, malocclusion, and complex orthodontic issues that may require extensive treatment later [1,2]. Crowns are one of the preferred treatment options for restoring function and esthetics in primary teeth, especially in cases of severe caries, developmental defects, or after pulp therapy [3].

The primary indications for using crowns in pediatric dentistry include extensive carious lesions that cannot be restored with conventional fillings, developmental anomalies such as hypoplasia, teeth that have undergone pulpotomy or pulpectomy, and structurally compromised teeth. Crowns are contraindicated in cases where there is insufficient remaining tooth structure for retention, severe periodontal involvement, or when the child’s behavior prevents effective treatment [3,4].

Pediatric crowns offer several advantages for primary teeth, including superior strength, durability, and complete coverage, which provides enhanced protection against further caries. They also help maintain occlusal function and prevent space loss, which is critical for the proper eruption of permanent teeth. Indeed, they enable effective and durable therapeutic outcomes, making them a favorable option for restoring primary teeth. Compared to traditional manual restorations, they often allow for quicker and more efficient treatment, particularly in cases involving extensive multi-surface carious lesions [3,4,5]. However, there are some limitations. Although stainless steel crowns (SSCs) are the first option for crown restorations on primary teeth, they may not provide esthetic appeal, and inadequately contoured crowns are the reason for gingivitis associated with SSCs. These crowns contain very low levels of nickel and are associated with a low risk of nickel allergy and hypersensitivity [3,5].

Recent advancements in technology and the increasing demand from both parents and children for more esthetic solutions have shifted the focus toward alternative materials instead of SSCs [6,7]. Prefabricated zirconia crowns have been developed to meet these esthetic expectations while providing high strength and durability for both anterior and posterior primary teeth [8,9,10]. However, prefabricated zirconia crowns have drawbacks, such as the inability to be modified and the need for passive seating during cementation, which may require more aggressive tooth preparation. They also cannot withstand flexure, increasing the risk of fracture during cementation, and if any part fractures, the entire crown must be replaced [11,12]. Laser-sintered, pre-cured nano-hybrid composite crowns have recently been introduced and expand the range of prefabricated pediatric crowns. Designed to replicate natural primary teeth, they offer excellent physical and esthetic properties for both anterior and posterior use. Cured through light, heat, or pressure, these crowns have homogeneous, non-porous surfaces, improving wear resistance and ensuring proper adaptation [13].

The use of computer-aided design and computer-aided manufacturing (CAD-CAM) in dentistry has advanced, utilizing either subtractive manufacturing with milling technology or additive manufacturing through three-dimensional (3D) printing [14,15]. CAD-CAM systems have enabled the production of customized esthetic and functional crowns for pediatric patients. Milling-based systems can create crowns from materials such as zirconia ceramics, resin-based ceramics, and composite resins [16,17,18]. Resin ceramic and composite blocks are ideal for primary teeth, as their lower hardness reduces wear on opposing teeth, and their low modulus of elasticity helps absorb functional stresses [16,17,18,19,20]. The introduction of 3D-printed crowns has opened new possibilities for dental restorations and is becoming a preferred alternative in pediatric dentistry [21,22]. The key advantages of 3D-printing technology include the ability to produce multiple objects in a single process, the reproduction of complex geometries, and its cost effectiveness [22,23]. The development of 3D-printing technology has led to the introduction of various innovative materials in pediatric dentistry. These materials include resin-based composites and high-performance polymers such as polyetheretherketone (PEEK). Resin-based materials are preferred for their esthetic properties, ease of customization, and elasticity moduli that closely resemble those of natural dentin, making them suitable for both anterior and posterior restorations [23,24]. Permanent crown resins from different companies for 3D printers have been introduced to the market for mid-to-long-term use in the oral cavity [25,26,27]. The successful use of these permanent resins in pediatric patients has been documented in a case report, which restored a severely damaged premolar tooth using a 3D-printed endocrown following coronal pulpotomy [28]. PEEK, a high-performance polymer, stands out for its superior biocompatibility and mechanical properties comparable to natural dentin, making it a promising material for long-term restorations [24,29]. The versatility of these 3D-printing materials allows for more precise and conservative tooth preparation, ultimately improving treatment outcomes in pediatric dentistry. Due to the wide variety of restorative materials, selecting the most suitable material for a given case can often be challenging for pediatric dentists. Factors such as enamel wear properties and fracture resistance, which can impact the long-term success of the restoration, play a crucial role in the selection of restorative materials for pediatric patients [13,30,31].

Therefore, the purpose of this study was to assess crown restorations in pediatric dentistry by utilizing the latest technology and materials that address the limitations of existing materials used in the restoration of primary molars. For this purpose, the wear properties of the primary tooth enamel and fracture resistances of different esthetic pediatric crowns used in primary molars were compared. The null hypothesis was that the type of restorative material would not impact the wear of primary teeth and fracture resistance of the restoration.

## 2. Materials and Methods

Ethics approval for this study was obtained from the Clinical Research Ethics Committee of Gazi University, Faculty of Dentistry, under approval number 2023.18/01 and date 21 September 2023.

### 2.1. Sample Size Calculation

The required sample size for this test was calculated using the “pwr” package in the R version 3.1.9.4. programming language. The effect size for comparing the groups in terms of wear was calculated as 0.61 using the mean and standard deviation values obtained from previous studies [29,32,33,34]. Based on this effect size, with a Type I error rate of 0.05 and a power of 0.95, it was calculated that 60 samples (at least 10 per group) were required for the study. For the compared groups in terms of fracture resistance, the effect size was calculated as 0.8 using the mean and standard deviation values obtained from previous studies [19,35,36]. Based on this effect size, with a Type I error rate of 0.05 and a power of 0.95, the minimum sample size per group was calculated as 7.

### 2.2. Study Design

Six different experimental groups were generated using restorative materials for pediatric crowns (Table 1). The mandibular second primary molar (85) on a pediatric dental model (AK-6/2M, Frasaco, Germany) was prepared with a 1 mm reduction on the axial surfaces, 1.5 mm on the occlusal surface, and a 1 mm chamfer margin. Sharp angles and edges were rounded. After tooth preparation, gypsum study models were fabricated to replicate the prepared tooth accurately. These models were then positioned and secured on a semi-adjustable articulator (Stratos 100, Ivoclar Vivadent, Schaan, Liechtenstein), simulating the natural occlusal relationships for further procedures. Sixty 3D-printed resin dies (n = 10) were then produced and embedded vertically in cylindrical molds with self-curing acrylic resin. Prefabricated zirconia and composite crowns of appropriate sizes were selected by fitting them on the prepared tooth.

### 2.3. Crown Manufacturing and Cementation

For milled and 3D-printed crowns, the digital model was transferred to the design software (DentalCad^®^, Exocad GmbH, Darmstadt, Germany). On the digital model, margins were defined, the cement space was set to 50 microns, and the crowns were designed to match the anatomical structure. The designs of the crowns were saved in STL file format. The composite block (Tetric CAD, Ivoclar Vivadent, Schaan, Liechtenstein), nanoceramic block (GC Cerasmart, GC Corporation, Tokyo, Japan), and PEEK (White Peaks Dental Systems GmbH, Essen, Germany) disc were milled using a milling unit (Redon, Hybrid Dental CNC, Redon Technology, İstanbul, Türkiye). For the 3D-printed crown production, the STL files were transferred to PreForm software (Formlabs Inc., Somerville, MA, USA), where the printing parameters were set and the file converted to a “.form” file. The crowns were printed using a Formlabs Form 3 printer with permanent crown resin (Formlabs, Somerville, MA, USA). After printing, the crowns were washed in 95% isopropyl alcohol for 3 min using the Form Wash&Cure unit (Formlabs Inc., Somerville, MA, USA), dried with compressed air, and then cured for 20 min to ensure dimensional stability. All produced specimens were mechanically polished (OptraPOL; Ivoclar Vivadent).

Before cementation, the 3D-printed resin dies were sandblasted with 50 µm Al_2_O_3_ particles at 2 bar pressure from a distance of 10 mm for 20 s. Subsequently, they were cleaned in an ultrasonic cleaner and air-dried. A bonding agent (Scotchbond Universal Plus; 3M ESPE, Seefeld, Germany) was applied to the resin dies for 20 s, dried with air, and surface contamination was prevented. Additional procedures were performed on the inner surface of the restorations according to the manufacturers’ recommendations for cementation (Table 2). All crowns were cemented using resin cement (RelyX Universal Resin Cement, 3M ESPE, Seefeld, Germany). The crowns were placed onto the resin dies under finger pressure. The crowns were then cured for 2–3 s using an LED curing device (Valo Grand; Ultradent Products, South Jordan, UT, USA). After removing excess cement, the crowns were cured for 10 s from each side of the crown. The cementation process was completed by waiting 5 min (Figure 1).

### 2.4. Wear Evaluation

For evaluating the amount of wear of the crowns on opposing primary molar enamel, 60 caries-free maxillary primary second molar teeth (55) were used. Teeth extracted from children whose parents gave verbal consent and signed an informed consent form before the procedure at the Pediatric Dentistry Clinic between February 2024 and May 2024 were included in the study. Teeth were selected based on the following inclusion criteria: absence of caries, structural defects, or abnormal morphology. The exclusion criteria included teeth with extensive caries, fractures, or teeth showing abnormal wear patterns. The teeth were extracted for orthodontic reasons, pathological causes, or physiological root resorption. The roots of the extracted teeth were retained only to the extent necessary to ensure proper retention during embedding in acrylic resin. The extracted teeth were cleaned and disinfected using a 0.5% chloramine-T solution for 48 hours and then stored in distilled water at room temperature for further use. These teeth were mounted on the loading jigs of the chewing simulator (Esetron Robotechnologies, Ankara, Türkiye). The specimens (teeth with crowns and natural antagonist upper teeth) underwent thermodynamic aging with mechanical loading in a dynamic loading device (chewing simulator) at 50 N and a frequency of 1.6 Hz for 250,000 cycles (approximately equivalent to one year of aging) [7,12]. The loading was applied with the palatal cusp tips of the maxillary primary second molar teeth in contact with the occlusal central fossa of the crowns. Simultaneously, the specimens were subjected to thermal cycling between temperatures of 50 °C and 55 °C.

All tooth specimens were scanned with a 3D intraoral dental scanner (Trios 5, 3Shape, Copenhagen, Denmark) before and after the wear test, and the scans were saved in STL file format (Figure 2). For vertical (2D) and volumetric (3D) wear measurements, the STL files obtained from the scans were imported into the software (3-matic 18.0, Mimics Innovation Suite 26.0, Materialise, Leuven, Belgium). The aged models were subtracted using a digital 3D superimposition technique. The STL files of the specimens before and after aging were aligned using the 3-matic software (Materialise, Belgium). The non-aged models were set as the reference, and the aged models were superimposed on them. The files were aligned, and after alignment, wear models were created by subtracting the aged models from the non-aged ones in their aligned positions (Figure 2). A subtraction algorithm within the software was then applied to calculate the volumetric (3D) and vertical (2D) differences between the two models, thereby quantifying the amount of wear. Vertical wear of maxillary primary second molar teeth was recorded in millimeters (mm), and volume loss was recorded in cubic millimeters (mm^3^).

### 2.5. Fracture Resistance Test

After thermomechanical aging, the teeth and crowns were inspected, and any cracks, micro-perforations, fractures, or losses were excluded. Vertical loading was then applied using a universal testing machine (LLYOD, Lloyd Instruments Ltd., Fareham, UK) at a speed of 1 mm/min with a 5 mm diameter ball until fracture occurred. The fracture resistance values were automatically recorded as the maximum loading force (N), and each specimen was categorized according to the types of failures (Table 3) [16].

### 2.6. Statistical Analyses

Statistical analyses were performed using SPSS software (version 20.0). The Shapiro–Wilk test was used to assess data normality, while Levene test was used to check for homogeneity of variances. One-way ANOVA was used to analyze both 2D and 3D wear data, which were normally distributed. Due to the non-homogeneity of variances in the 2D wear data (*p* = 0.003), Welch’s correction was applied. Fracture resistance data also showed normal distribution (*p* > 0.05), and one-way ANOVA was used; however, the non-homogeneity of variances (*p* < 0.001) required Welch’s correction and the Games–Howell test for pairwise comparisons. The significance level was set at α = 0.05.

## 3. Results

The descriptive statistics for the 3D and 2D wear data of the experimental groups are shown in Table 4. The highest amount of 3D wear was found in Group 3 (milled composite crown). The descriptive statistics for the 3D and 2D wear data of the experimental groups are shown in Table 4. The highest amount of 3D wear was found in Group 3 (milled composite crown) at 1.74 ± 0.95 mm^3^, while the lowest wear amount was recorded in Group 4 (milled resin matrix ceramic crown) at 0.99 ± 0.66 mm^3^. The highest 2D wear amount was found in Group 6 (3D-printed crown) at 1.01 ± 0.43 mm, while the lowest wear amount was recorded in Group 1 (prefabricated zirconia crown) at 0.59 ± 0.23 mm. ANOVA results showed that the type of restoration used in the study did not significantly affect the 3D (*p* = 0.270) and 2D (*p* = 0.381; Welch: 0.095) wear amounts of primary teeth (Table 5).

According to the ANOVA test results, the type of restorative material had a significant effect on the fracture resistance of pediatric crowns (*p* < 0.001) (Table 6). The significantly highest fracture resistance was found in Group 5 (PEEK crown) at 1797 ± 412.6 N (*p* < 0.05). The significantly lowest fracture resistance was recorded in Group 1 (prefabricated zirconia crown) at 521.4 ± 63.16 N (*p* < 0.05) (Table 7).

During thermomechanical aging, no failures were observed in any of the specimens. All specimens were subjected to fracture testing, and the failure types for the experimental groups are shown in Table 8 and Figure 3. All failure types were visible to the naked eye.

## 4. Discussion

The purpose of this study was to determine and compare the amount of enamel wear on primary teeth and the fracture resistance of materials used for pediatric crowns. The null hypothesis of the study was that the restorative material type would not affect the wear of primary tooth and fracture resistance. According to the results, the type of restorative material did not significantly affect the amount of wear on primary teeth. However, the type of restorative material had a significant effect on the fracture resistance of pediatric crowns. Thus, the null hypothesis was partially rejected.

Several studies have evaluated the mechanical properties of prefabricated zirconia and composite crowns for primary teeth, often comparing different types of prefabricated crowns or SSCs [6,7,16,17,18,19,20,21]. However, digital dentistry has become increasingly integrated into pediatric dental practices, with CAD-CAM systems allowing for highly accurate and personalized restorations [22]. However, studies do not comprehensively evaluate all crown options for primary molars, especially with the latest esthetic materials and techniques. For this reason, in the present study, prefabricated zirconia crowns (NuSmile), composite resin crowns (Edelweiss), milledPEEK, resin matrix ceramic and composite resin crowns, and 3D-printed permanent composite resin crowns were tested in terms of wear and fracture resistance after thermomechanical aging. Chieruzzi et al. [37] emphasized that the strength of dental systems under masticatory loads is closely related to the bond integrity at the post/cement and cement/dentine interfaces. A strong bond at these structures ensures better stress distribution, reducing the risk of debonding or failure [37]. Material selection, surface treatment, and proper bonding protocols play crucial roles in optimizing the bond between structures, while factors such as aging and cyclic loading can degrade it over time [37,38]. In the present study, to obtain standardized specimens, the pediatric crowns were cemented on the standardized resin dies by using resin cement.

Recent studies have employed 3D evaluation methods to assess the wear behavior of dental materials [39,40,41]. In the present study, wear measurement involved 3D digital scans taken before and after wear testing, with 2D vertical loss (mm) and 3D volume loss (mm^3^) calculated by overlaying and subtracting the two 3D models. This quantitative methodology is highly recommended by many authors due to its accuracy [42,43].

In the present study, the type of restoration did not significantly affect the 3D and 2D wear amounts of the primary teeth. The highest 3D wear amount was in the milled resin matrix ceramic crown group. Möhn et al. [20] evaluated the wear properties of stainless steel, zirconia, resin composite, and hybrid ceramic crowns on antagonistic teeth and found that hybrid ceramic crowns exhibited lower wear depth compared to zirconia crowns. Nakase et al. [44] reported that milled resin composite materials caused less wear on antagonist teeth compared to hybrid ceramic and composite materials. The differences in results are likely due to varying protocols in thermomechanical aging. In the present study, enamel wear was found to be lower in the zirconia group, consistent with previous findings [6,9,13]. Walia et al. [45] reported minimal enamel loss on opposing primary incisors in contact with zirconia crowns and no surface changes with composite resin crowns. It was also reported that smooth surfaces of restorations can reduce the wear rate of opposing teeth [6]. In addition, Donovan et al. [46] noted that only glazed or poorly polished zirconia could damage opposing teeth. However, controversial results were also reported. Taran et al. [12] reported that milled zirconia crowns caused the most enamel wear in opposing teeth. Talekar et al. [30] found that zirconia crowns caused significantly more wear on opposing primary canine teeth compared to glass-fiber-reinforced composite crowns. In this study, 3D-printed crowns showed comparable 2D and 3D wear to other tested groups.

Recently, different companies have introduced permanent crown resins for 3D printers, and these are now being used in permanent tooth restorations [25,47,48]. The tested 3D-printed resin was a permanent crown resin by Formlabs, which is thought to be more wear-resistant due to its material properties. In the present study, the 3D wear volume of prefabricated composite crowns was measured as 1.62 ± 1.38 mm^3^. Signoriello et al. [13] evaluated the wear properties of zirconia, nano-hybrid composite (Edelweiss), and stainless steel crowns on opposing primary teeth. Zirconia crowns showed the smallest wear (0.98 mm²), while nano-hybrid composite crowns exhibited the largest wear area (5.6 mm²). The difference in wear could be attributed to variations in filler content and composition of the composite materials. Additionally, the type of opposing teeth and testing protocols used in the study may have contributed to the differences in wear characteristics. PEEK crowns caused less wear on opposing teeth in this study, though not significantly different from other groups. Choi et al. [29] reported similar findings with high-performance polymers. Few studies specifically examine the use of PEEK in pediatric crowns for primary tooth wear.

The fracture resistance of the restorative materials should be higher than the masticatory forces. Studies have reported varying maximum bite forces in children [48,49,50]. Kamegai et al. [48] found that children aged 6–11 in northern Japan had an average bite force of 374.4 to 433 N. Braun et al. [49] reported an average bite force of 106 N in children aged 6–12, while Owais et al. [50] observed a maximum bite force of 433 N during the mixed dentition period. In the present study, it was found that PEEK crowns exhibited the highest fracture resistance among the tested groups, while prefabricated zirconia crowns exhibited the lowest. At the same time, no specific studies on pediatric PEEK crowns were found. Lu et al. [51] demonstrated high fracture loads for PEEK crowns in permanent teeth restorations. The results from our study align with the findings of the previous studies [7,16,17]. In the present study, no significant difference in fracture resistance was observed among prefabricated composite, milled composite (643.80 ± 96.61 N), milled resin matrix ceramic (596 ± 78.54 N), and 3D-printed crowns (625.2 ± 105.62 N). The variations in fracture resistance values across studies could be attributed to differences in material compositions, manufacturing processes, types of cement used, and testing protocols. Additionally, composite materials’ tendency to absorb water or hydrolyze silanes in a moist environment may result in lower wear and fracture resistance [52]. To the best of our knowledge, there are limited studies regarding prefabricated composite resin crowns. Our findings suggest that tested prefabricated resin crowns can withstand chewing forces effectively, making them a viable option for pediatric dentistry. Although prefabricated zirconia crowns showed the lowest fracture resistance value in the present study, it was observed that all tested crowns exhibited significantly higher fracture resistance than the reported average bite forces in children [48,49,50]. This suggests that the tested crowns possess mechanical properties suitable for withstanding chewing forces and can be safely used in pediatric dentistry.

This study has some limitations. The study included six different types of restorations used in pediatric crowns. However, many other materials can be used for pediatric crowns in CAD-CAM systems. Additionally, different 3D-printing techniques can also be employed for crown production. There is a need for further testing of pediatric crowns produced using various CAD-CAM materials and 3D-printing technologies under aging protocols with longer cycles. In this study, the crowns were cemented onto standardized resin dies, and the bonding mechanism of the resin die differs from that of primary teeth. Although in vitro studies provide promising results, clinical studies are necessary to determine the most suitable crown materials for pediatric crowns.

## 5. Conclusions

Within the limitations of this in vitro study, the following conclusions are drawn:The restoration type did not have a significant effect on the 2D and 3D wear of the primary tooth enamel.The fracture resistance of the tested materials differed according to the material type. Although the milled PEEK group showed the highest fracture resistance, all tested materials could withstand chewing forces in children.

## Figures and Tables

**Figure 1 medicina-60-01678-f001:**
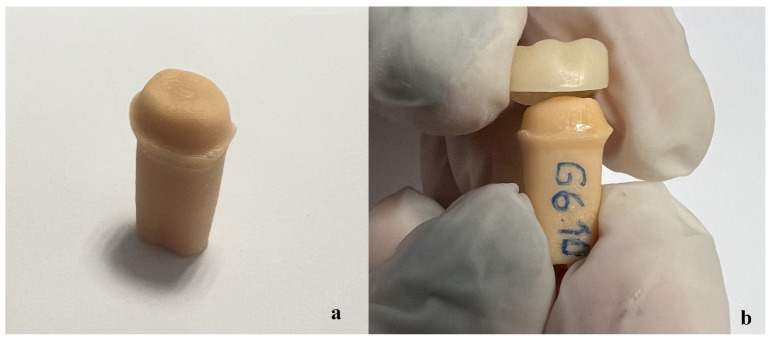
(**a**) The 3D-printed resin die. (**b**) Cementation of the crown onto the resin die with resin cement.

**Figure 2 medicina-60-01678-f002:**
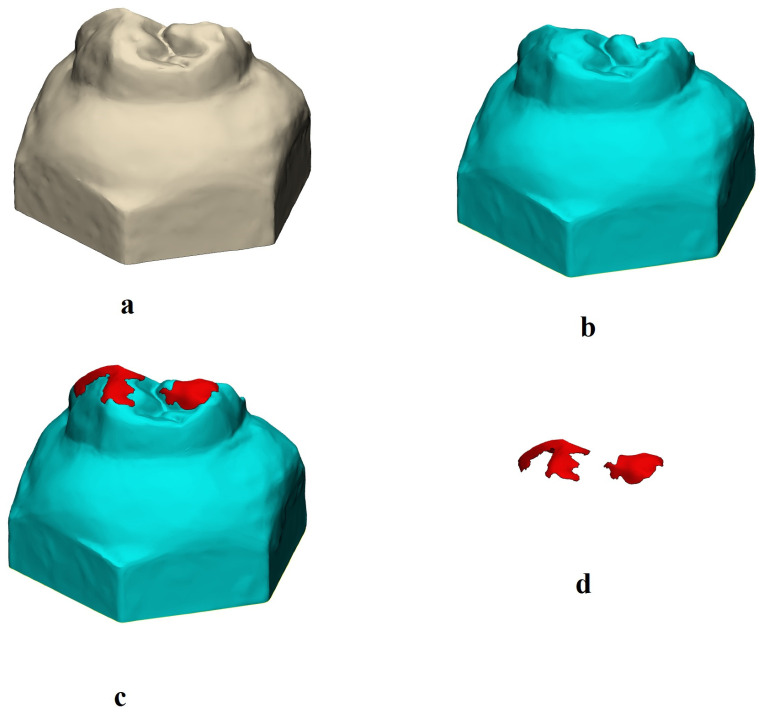
(**a**) STL files of tooth specimens before the wear test, (**b**) STL files of the same tooth specimens after the test, (**c**) overlapping of the STL files of the tooth specimens before and after testing, and (**d**) STL files of the worn region.

**Figure 3 medicina-60-01678-f003:**
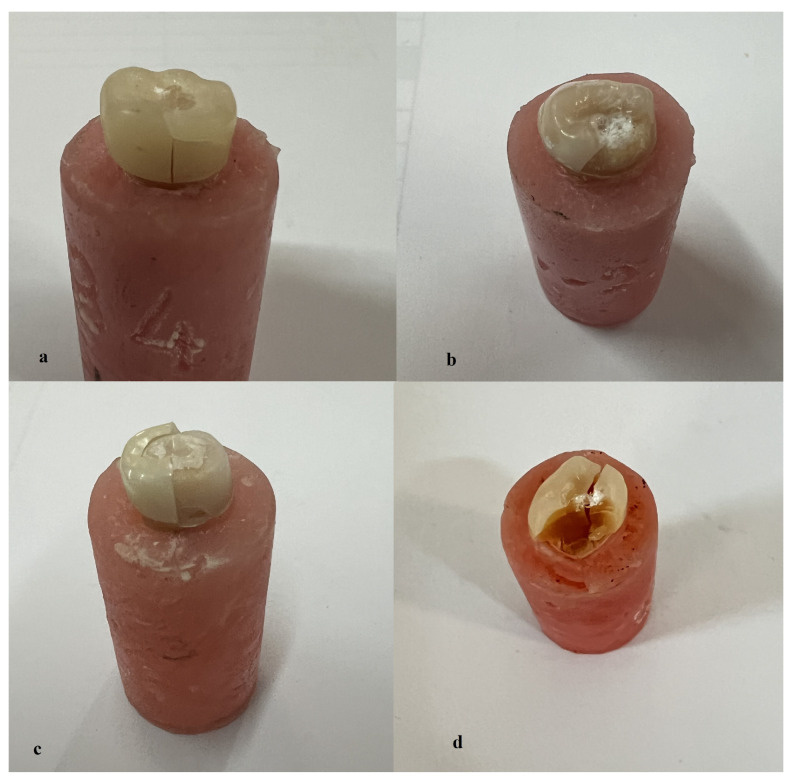
Examples of failure types for experimental groups: (**a**) Type III failure; (**b**) Type IV failure; (**c**) Type V failure; (**d**) Type VI failure.

**Table 1 medicina-60-01678-t001:** Materials used in the study.

Group/Material	Composition	Manufacturer	Production Method
**Group 1:** Prefabricated Zirconia Crown	88–96% zirconium oxide, 4–6% yttrium oxide, 5% hafnium oxide, 2–5% organic binders, 1–4% pigments	NuSmile, Houston, TX, USA	Prefabricated Crown
**Group 2:** Prefabricated Composite Crown	82% inorganic filler (particle size 0.02–3 μm) (barium glass, Bis-GMA-based matrix, pigments, additives, catalysts)	Edelweiss, Wolfurt, Austria	Prefabricated Crown
**Group 3:** Composite Crown	28.4% cross-linked dimethacrylates, 71.1% barium glass, silicon dioxide	Ivoclar Vivadent, Schaan, Liechtenstein	Subtractive Manufacturing (CAD-CAM; milling)
**Group 4:** Resin Matrix Ceramic Crown	29% Bis-MEPP, UDMA, DMA, 71% silica (20 nm), barium glass (300 nm) nanoparticles	GC Corporation, Tokyo, Japan	Subtractive Manufacturing (CAD-CAM; milling)
**Group 5:** PEEK Crown	100% polyetheretherketone (PEEK)	White Peaks Dental Systems GmbH, Essen, Germany	Subtractive Manufacturing (CAD-CAM; milling)
**Group 6:** Permanent Crown Resin	Organic matrix: 50– < 75% wt. Bis-EMA Esterification products of 4.4′-isopropylidiphenol, ethoxylated and 2-methylprop-2enoic acid. Silanized dental glass, methyl benzoylformate, diphenyl [2,4,6-trimethylbenzoyl] phosphine oxide. Inorganic filler: Silanized dental glass (particle size 0.7 μm) (30–50% wt.)	Formlabs Inc., Somerville, MA, USA	Additive Manufacturing (3D-printing)

**Table 2 medicina-60-01678-t002:** Procedures for cementation of crowns used in the study.

Group/Material	Preparation of the Internal Surface before Cementation
**Group 1:** Prefabricated Zirconia Crown	No preparation is required
**Group 2:** Prefabricated Composite Crown	Sandblasting with Al_2_O_3_ particles Cleaning with ethanol in an ultrasonic bath Rinsing and drying with air Application of bonding agent (3M Scotchbond) Drying with air
**Group 3:** Composite Crown	Sandblasting with Al_2_O_3_ particles Cleaning with ethanol in an ultrasonic bath Rinsing and drying with air Application of bonding agent (3M Scotchbond) Drying with air
**Group 4:** Resin Matrix Ceramic Crown	Application of 5% hydrofluoric acid (IPS Ceramic Etching Gel; Ivoclar Vivadent) for 60 s Application of bonding agent (3M Scotchbond)
**Group 5:** PEEK Crown	Sandblasting with Al_2_O_3_ particles Application of bonding agent (3M Scotchbond) Drying with air
**Group 6:** Permanent Crown Resin	Sandblasting with Al_2_O_3_ particles Cleaning with ethanol in an ultrasonic bath Rinsing and drying with air Application of bonding agent (3M Scotchbond) Drying with air

**Table 3 medicina-60-01678-t003:** Types of failures and their definitions.

Failure Type	Definitions
**I**	Cracks that are not visible to the naked eye but can be seen under a stereomicroscope
**II**	Visible cracks on unseparated margins
**III**	Cracks on separated margins
**IV**	Crown fracture with less than half of the crown displaced, with the supporting structure intact
**V**	Crown fracture with more than half of the crown displaced, with the supporting structure intact
**VI**	Crown fracture involving the supporting structure

**Table 4 medicina-60-01678-t004:** The descriptive statistics for the 3D and 2D wear data of the experimental groups.

Material (n = 10)	Mean	Standard Deviation	Standard Error	95% Confidence Interval		Minimum Value	Maximum Value
				Upper Limit	Lower Limit		
**3D Wear**							
**Group 1:** Prefabricated Zirconia Crown	1.69	0.74	0.23	1.16	2.22	0.30	2.8
**Group 2:** Prefabricated Composite Crown	1.62	1.38	0.44	0.63	2.61	0.00	3.9
**Group 3:** Composite Crown	1.74	0.95	0.3	1.06	2.42	0.10	3.5
**Group 4:** Resin Matrix Ceramic Crown	0.99	0.66	0.21	0.52	1.47	0.10	2.2
**Group 5:** PEEK Crown	1.1	0.49	0.16	0.75	1.45	0.30	2.2
**Group 6:** Permanent Crown Resin	1.29	0.82	0.26	0.71	1.88	0.10	3.2
Total	1.41	0.9	0.11	1.17	1.64	0.00	3.9
**2D Wear**							
**Group 1:** Prefabricated Zirconia Crown	0.59	0.23	0.07	0.43	0.75	0.2	1
**Group 2:** Prefabricated Composite Crown	0.97	0.53	0.17	0.59	1.35	0.1	1.7
**Group 3:** Composite Crown	0.88	0.34	0.11	0.64	1.13	0.2	1.5
**Group 4:** Resin Matrix Ceramic Crown	0.84	0.75	0.24	0.3	1.38	0.1	2.1
**Group 5:** PEEK Crown	0.73	0.39	0.12	0.45	1	0.02	1.5
**Group 6:** Permanent Crown Resin	1.01	0.43	0.17	0.7	1.32	0.4	1.8
Total	0.84	0.48	0.061	0.7	0.96	0.02	2.1

**Table 5 medicina-60-01678-t005:** One-way ANOVA results of 3D and 2D wear amounts of primary teeth.

Source of Variation	Sum of Squares	Degrees of Freedom	Mean Squares	F Value	*p* Value
**3D Wear**					
Between Groups	5.181	5	1.036	1.318	0.270
Within Groups	42.467	54	0.786		
Total	47.649	5			
**2D Wear**					
Between Groups	1.215	5	0.243	1.082	0.381
Within Groups	12.131	54	0.225		
**Total**	13.346	59			

**Table 6 medicina-60-01678-t006:** One-way ANOVA results of the fracture resistance.

Source of Variation	Sum of Squares	Degrees of Freedom	Mean Squares	F Value	*p* Value
Between Groups	11,958,033.950	5	2,391,606.790	67.797	0.000
Within Groups	1,904,895.700	54	35,275.846		
Total	13,862,929.650	59			

**Table 7 medicina-60-01678-t007:** The descriptive statistics for the fracture resistance.

Material (n = 10)	Mean	Standard Deviation	Standard Error	95% Confidence Interval		Minimum Value	Maximum Value
				Upper Limit	Lower LIMIT		
**Group 1:** **Prefabricated Zirconia Crown**	**521.40 C**	**63.16**	**19.97**	**476.22**	566.58	430	645
**Group 2:** Prefabricated Composite Crown	633.70 **BC**	103.79	32.82	559.45	707.95	428	772
**Group 3:** Composite Crown	643.80 **B**	96.61	30.55	574.69	712.91	497	793
**Group 4:** Resin Matrix Ceramic Crown	596.00 **BC**	78.54	24.84	539.82	652.18	465	757
**Group 5:** PEEK Crown	1797.00 **A**	412.60	130.47	1501.85	2092.15	1315	2684
**Group 6:** Permanent Crown Resin	625.20 **BC**	105.62	33.40	549.64	700.76	461	835
Total	802.85	484.73	62.58	677.63	928.07	428	2684

The same capital letters indicate that the fracture resistance values of the experimental groups were not significantly different (*p* < 0.05).

**Table 8 medicina-60-01678-t008:** The failure types for the experimental groups.

Failure Type/Crown Group	Type I	Type II	Type III	Type IV	Type V	Type VI
**Group 1:** Prefabricated Zirconia Crown	-	-	-	-	8	2
**Group 2:** Prefabricated Composite Crown	-	-	-	2	-	8
**Group 3:** Composite Crown	-	-	-	-	-	10
**Group 4:** Resin Matrix Ceramic Crown	-	-	-	-	-	10
**Group 5:** PEEK Crown	-	-	-	-	-	10
**Group 6:** Permanent Crown Resin	-	-	1	-	-	9

## Data Availability

The datasets used and analyzed during the present study are available from the corresponding author upon reasonable request.

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
