# Peer review of "Evaluation of Wear on Primary Tooth Enamel and Fracture Resistance of Esthetic Pediatric Crowns Manufactured from Different Materials"

_medicina, 2024, doi:10.3390/medicina60101678_

Round 1

Reviewer 1 Report

Comments and Suggestions for Authors

Well structured article on new 3d materials for the fabrication of crowns in infant dentition. Only some criticisms listed below

-In the abstract section remove all commercial names of the materials tested

-check that all keywords are Pubmed MESH terms

-in the introduction section some more detailed considerations on the types of 3d printing materials introduced on the market, to make the description more detailed

-define why these materials were chosen in the study

-in fact from the table it appears that only one is with additive technique, why?

-some considerations on the relationship between tooth structure and resistance to masticatory loads should be carried out. In this regard I recommend inserting the following scientific work in the reference section that could be of help to the reader:

Chieruzzi M, Rallini M, Pagano S, Eramo S, D'Errico P, Torre L, Kenny JM. Mechanical effect of static loading on endodontically treated teeth restored with fiber-reinforced posts. J Biomed Mater Res B Appl Biomater. 2014 Feb;102(2):384-94. doi: 10.1002/jbm.b.33017. Epub 2013 Sep 2. PMID: 24000235.

-I request a total correction of the work as English language style

Comments on the Quality of English Language

low quality of english language

Author Response

Dear reviewer,

Please see the attchment. 

Reviewer 2 Report

Comments and Suggestions for Authors

This paper presents an intriguing contribution that addresses a significant gap in the existing literature regarding the wear of crowns on opposing primary molar enamel. While the study offers valuable insights, there are several areas where it could be improved. Notably, the absence of a sample size analysis may affect the robustness of the findings, and the paper could benefit from graphical representations of data rather than relying solely on SPSS output tables. Additionally, the paper mentions using 60 caries-free maxillary primary second molar teeth but does not clarify the reasons for extraction or whether any root structure remained, which could influence the results. Moreover, Table 7 includes abbreviations (A, B, and C) that are not explained in the table description, potentially confusing readers. The study also lacks a discussion on its limitations and does not provide a concluding section, both of which are essential for contextualizing the findings and guiding future research. Addressing these issues would greatly enhance the paper’s clarity and impact.

Author Response

Dear reviewer, 

Reviewer 3 Report

Comments and Suggestions for Authors

Dear Authors,

This study addresses a contribution to dental research by focusing on different primary crowns on temporary teeth. This focus on technology that is not widely studied offers unique contributions to pediatric dentistry.

I noted the complex methodology used to test the different pediatric crowns, but the methodology is not clearly described.

Abstract 

The conclusion must be clarified, and the study's aim must be answered because it refers mainly to the used restorations, not to the enamel wear, which is stated in the study's aim and the title of the manuscript.

Introduction

Authors should include the indications, contraindications, and advantages of crowns in primary teeth.

More information needs to be provided about the importance of using these kinds of restorations in pediatric dentistry and the complications that can occur if they are not used. What situations require these kinds of treatments?

Methods

The authors should include in the text the Ethical approval number, not only in the Institutional Review Board Statement section.

How was the sample size calculated?

The reporting methodology's sequence needs to be clarified. Please rearrange this section with subheadings in sequence, such as study design/tooth preparation (inclusion and exclusion criteria of the collected teeth; the reason for tooth extraction should be mentioned; manipulation of the teeth—disinfection, preparation, impressions) / crown manufacturing and cementation/wear evaluation/fracture resistance test / statistical analysis.

Pre-prepared teeth could have been used for standardization, as hand preparation of teeth introduces factors that could negatively impact the prosthodontic results. How invasive was the tooth preparation and how it was standardized?

Line 77-78: The methodology is not clear, please clarify.

Line 105: The finger pressure can be quantified by using a balance to standardize.

Line 117: How were the extracted teeth stored and prepared for the study?

Line 129: Which specimens? Specify: teeth with crowns, antagonist upper teeth?

Line 133: What kind of alignment was used?

Line 134: How were the aged models subtracted?

Line 135: On which teeth were the vertical wear recorded?

The figures should be organized according to the journal's recommendations. Please rearrange.

Figures 2 and 3 should be bigger to be easier to visualize. Please correct the numbering of Figure 3.

Results

The results are described in detail.

Discussion

The Discussion section should be separated into paragraphs based on the study findings and sequences.

What are the potential advantages and disadvantages of using these restorations? This should be included in the Introduction to clearly establish the research rationale. The same should be done in the Discussion section as well.

The study's limitations are missing, and future directions in the clinical application of these restorations are recommended.

Conclusions

The conclusions need to be included in the text. Must be in accordance with the research hypothesis and answer to the aim of the study.

All the cited references are relevant to the research.

Author Response

Dear reviewer, 

Round 2

Reviewer 1 Report

Comments and Suggestions for Authors

all questions were added

Reviewer 2 Report

Comments and Suggestions for Authors

The manuscript is greatly improved. I believe it is suitable for publication.
Congrats!